

**Equatorial Plasma Bubbles Developing Around Sunrise**
**Observed by an All-Sky Imager and GNSS Network during**
**the Storm Time**
Kun Wu[1,2], Jiyao Xu[1,2], Xinan Yue[3,2], Chao Xiong[4], Wenbin Wang[6], Wei Yuan[1,2], Chi
Wang[1,2], Yajun Zhu[5], Ji Luo[1,2]
[1]State Key Laboratory of Space Weather, National Space Science Center, Chinese Academy of
Sciences, Beijing, China
[2]College of Earth and Planetary Sciences, University of Chinese Academy of Sciences, Beijing,
China
[3]Key Laboratory of Earth and Planetary Physics, Institute of Geology and Geophysics, Chinese
Academy of Sciences, Beijing, China
[4]GFZ German Research Centre for Geosciences, Telegrafenberg, 14473 Potsdam, Germany.
[5]Institute of Energy and Climate Research (IEK-7), Forschungszentrum Juelich GmbH, Juelich,
Germany
[6]High Altitude Observatory, National Center for Atmospheric Research, Boulder, CO, USA
Correspondence to: jyxu@spaceweather.ac.cn
*Keywords:* Equatorial plasma bubble near sunrise, Spread-F, All-sky imager, GNSS
network





**Abstract.**

A large number of studies have shown that equatorial plasma bubbles (EPBs) occur mainly after sunset, and they usually drift eastward. However, in this paper, an unusual EPB event was simultaneously observed by an all-sky imager and the Global Navigation Satellite Systems (GNSS) network in southern China, during the recovery phase of geomagnetic storm happened on 6-8 November 2015. Observations from both techniques show that the EPBs appeared near dawn. Interestingly, the observational results show that the EPBs continued to develop after sunrise, and disappeared about one hour after sunrise. The development stage of EPBs lasted for at least about 3 hours. To our knowledge, this is the first time that the evolution of EPBs developing around sunrise was observed by an all-sky imager and the GNSS network. Our observation showed that the EPBs drifted westward, which was different from the usually eastward drifts of post-sunset EPBs. The simulation from TIE-GCM model suggest that the westward drift of EPBs should be related to the enhanced westward winds at storm time. Besides, break and recombination processes of EPBs were observed by the all-sky imager in the event. Associated with the development of EPBs, increasing in the ionospheric F region peak height was also observed near sunrise, and we suggest the enhance upward vertical plasma drift during geomagnetic storm plays a major role in triggering the EPBs near sunrise.

## 1. Introduction

After sunset, plasma density depletions, also called equatorial plasma bubbles (EPBs), sometime occur in the equatorial- and low-latitude ionosphere. A large number of studies have shown that EPBs generally start to develop shortly after sunset during geomagnetic quiet periods (e.g., Weber et al., 1980; Kelley et al., 1986; Xiong et al., 2010; Wu et al., 2018). It is generally believed that the Rayleigh-Taylor instability (RTI) is a plausible mechanism to trigger the EPBs (Kelley, 2009; Makela and Otsuka, 2012). The growth rate of RTI is influenced by a number of different factors, such as the zonal electric field, neutral wind and the background ionospheric/thermosphere, as well as



the strength of magnetic fields (Ott, 1978; Abdu, 2001; Burke et al, 2004). The pre-
reversal enhancement (PRE) of the eastward electric field around sunset is a main
reason for the development of EPBs (e.g., Fejer et al., 1999; Abdu, 2001; Kelley, 2009;
Huang, 2018). Owning to the intensified eastward electric field, near magnetic equator
the ionosphere is rapidly elevated to higher altitudes via $E \times B$ drifts, which is favorable
for the growth of RTI at the bottomside of the ionosphere.
The EPBs are thought to extend along magnetic field lines, and can reach as high as
magnetic latitudes of about $\pm 20°$ (Kelley, 2009; Lühr et al., 2014). Xiong et al. (2016,
2018) suggest that EPBs have a typical zonal size of about 50 km, by using Swarm in
situ electron density measurements as well as ground-based airglow imager. Although
the characteristics of EPBs have been widely studied, special events, especially those
occurring during geomagnetic storms, are still one of the interesting issues to be fully
addressed. Some of the results showed that geomagnetic storms can affect the
development of EPBs (e.g., Abdu et al., 2003; Tulasi et al., 2008; Carter et al., 2016),
and in some extreme cases, the EPBs can extend to middle latitudes during intense
geomagnetic storms (e.g., Sahai et al., 2009; Patra et al., 2016; Katamzi-Joseph et al.,
2017; Aa et al., 2018). Moreover, in the storm time, EPBs near sunrise were
occasionally observed by some instruments such as radar and satellite. Fukao et al.
(2003) used observations from the Equatorial Atmosphere Radar to report EPBs near
sunrise over the Indonesian region during a geomagnetic storm and suggested that the
EPBs were likely associated with the geomagnetic storm. Huang et al. (2013) reported
the observations of long-lasting daytime EPBs with the Communications/Navigation
Outage Forecasting System (C/NOFS) satellite during a geomagnetic storm in which
the EPBs were persistent from the post-midnight sector through the afternoon sector.
Zhou et al. (2016) used observations from multiple low Earth orbiting satellites, like
the Swarm constellation, the Gravity Recovery and Climate Experiment (GRACE)
satellite, and the C/NOFS satellite, to detect the EPBs around sunrise during the St
Patrick's Day storm. They suggested that the geomagnetic storm induced changes in
ionospheric dynamics should be the reason for triggering the EPBs. But until now, there



has been no research on the occurrence characters and evolution of EPBs around sunrise
using optical remote sensing, which can provide different aspects of the EPBs near
sunrise.
It is well known that the EPBs usually drift eastward as reported by many studies (e.g.,
Pimenta et al., 2001; Martinis et al., 2003; Park et al., 2007; Taylor et al., 2013; Wu et
al., 2017). However, during storm periods westward drifting EPBs have been also
observed (Abdu et al., 2003; Basu et al., 2010; Santos et al., 2016). Abdu et al. (2003)
reported some cases of EPBs that showed eastward drifts after sunset and later reversed
to westward. Basu et al. (2010) reported that the westward drifting EPBs reached
maximum velocities of about 80 - 120 m/s. Santos et al. (2016) also showed some EPBs
of zonal drifts reversal (eastward to westward) during a geomagnetic storm, in which
they suggested the Hall electric field caused the reversal.
From six-year observations of airglow image located in the southern China, we found
only one case of EPBs starting to appear near sunrise during the storm recovery phase
on 08 November 2015. The EPBs appeared before sunrise, kept developing and
vanished in about 1 hour after sunrise. Unlike the quiet-time eastward drifting EPBs,
the EPBs drifted westward. In the rest, we provide a detailed analysis of this event. In
section 2, we give a general description of the instruments. Observational results are
showed in section 3. In section 4, we provide comparisons with previous studies as well
as discussions. Finally, summary is given in section 5.

**2.   Instrumentation**
**2.1     All-sky imager**
The airglow data used in this study are obtained from an all-sky imager, which is
deployed at Qujing, China (Geographic: 25° N, 104° E; Geomagnetic: 15.1° N, 176°
E). Its location is indicated by the red star in Figure 1, and the blue circle represents the
field of view (FOV) of the all-sky imager at an altitude of 250 km. The all-sky imager
consists of a CCD detector (1024 × 1024 pixel), an interference filter (630.0 nm), and
a fish-eye lens (FOV of 180°). The integration time of the all-sky imager is 3 min.




## 2.2 The Network of Global Navigation Satellite System (GNSS)

The GNSS data used in this study are derived from the Crustal Movement Observation Network of China (CMONOC), which consists of ~260 ground GNSS receivers covering the mainland of China. The information of these GNSS receivers has been given in previous publications (e.g., Aa et al., 2015; Yang et al., 2016; Zheng et al., 2016). The total electron content (TEC) was processed using the similar method as that described by Ding et al. (2014). Specifically, for each arc, the relative phase TEC was filtered using a band-pass filter. We then calculated the TEC residual of each arc for each pierce point, which the height of each ionospheric pierce point was about 300 km. Therefore, the TEC residual could indicate the occurrence of plasma bubbles. An elevation cutoff angle of 30° is used to reduce the multi-paths effects.

## 2.3 Digisond

The digisonde ionograms are obtained from a digisonde located at Fuke, a low-latitude station in the southern China (Geographic: 19.5° N, 109.1° E; Geomagnetic: 9.5° N, 178.4° W), and marked with a green dot in Figure 1. The virtual heights of the *F* layer were manually scaled by using the SAO Explorer software.

## 3. Observations and Results

Figure 2 shows the 3-hour *Kp* index, the interplanetary magnetic field (IMF) *Bz*, *SYM-H*, AE, AU, AL and h' F at Fuke on 06-08 November 2015. To make the comparison easier with other observations, we converted the universal time to the local time (LT) at Qujing. A geomagnetic storm occurred during those days. In Figure 2(b), IMF *Bz* turned southward at ~11:40 LT on 07 November 2015, and reached to about -11 nT at ~16:00 LT. During the storm main phase, the *SYM-H* had a rapid reduction from -40 nT to -100 nT. Meanwhile, the *Kp* index reached a value of 6; the AE and AL also reached at ~1500 nT and ~- 1500 nT, respectively. After 04:00 LT on 08 November 2015, IMF *Bz* began to turn to north. In the storm recovery phase, the value of *SYM-H* was back to



-40 nT.
Figure 3 shows the time sequence of airglow images observed by the all-sky imager at
Qujing from 05:15 to 06:21 LT on 8 November 2015. The time difference between
successive images is 6 min. For each image, we removed the effects of compression
and curving of the all-sky imager lens by an unwarping process (Garcia et al., 1997).
All images have been mapped into a geographic range from 97° to 111° E in longitude
and from 18° to 32° N in latitude. The height of the airglow layer is assumed to be at
250 km. The top of each image is to the north and the right to the east. Two EPBs,
marked as "b1" and "b2", were observed by the all-sky imager during this period. They
occurred during the geomagnetic storm recovery phase.
Around 05:21 LT, EPB "b1" appeared in the FOV of the all-sky imager. "b1" was still
developing, as it extended northward and reached close to 25° N around 06:21 LT. At
05:39 LT, the other EPB "b2" started to appear in the FOV of the airglow imager. "b2"
was also developing and expanded to about 20° N at 06:21 LT. The two observed EPBs
possibly continued to develop after 06:21 LT, as no hints of stop can be seen in the last
airglow image. However, there was no further image data after 06:21 LT because the
all-sky imager had to be shut down after sunrise. We want to pointed out that the sunrise
time at Qujing was around 06:15 LT at altitude of 250 km on that day. The far north
part of "b1" reached about 24.5°N at 06:15 LT. After 6 min, the far north of "b1"
extended to about 25°N (as marked by the black horizontal line). In other words, the
observational result from the all-sky imager suggested that the EPBs kept developing
after sunrise.
Some interesting features can also be seen from Figure 3. "b1" appeared at ~105° E and
"b2" appeared at ~104° E at 05:39 LT. Based on the black vertical line at 106° E, we
can clearly see that the two EPBs drifted from east to west. Besides, break and
recombination processes of EPB "b1" were also observed. After 05:45 LT, a break
process occurred in "b1". The lower latitude portion of "b1" moved further to the
westward. An obvious cleft occurred at ~19° N of "b1" near 06:03 LT. More interesting
is the fact that a recombination process occurred in the two break portions of "b1"



during its later development period. After ~06:03 LT, the upper portion of "b1" began
to connect to the lower portion of "b1" and they merged/combined together into one
EPB after 06:15 LT. The break and recombination processes are more obvious in the
red rectangles of Figure 3, which is indicated by the red arrow in each image.
Figure 4 shows a series of TEC residuals over 10°-50°N and 80°-130°E during 04:30-
08:20 LT on 08 November 2015. The adjacent imaging is in 10 min intervals. At about
04:40 LT, some TEC depletions, which occurred to the south and west of the location
of all-sky imager, appeared at ~115°E (~24°N), and began to develop. About 05:30 LT,
some additional EPBs appeared at ~105°E (~20°N), and they were also developing.
EPBs in the two regions kept developing until they disappeared. Owning to the FOV of
the all-sky imager, the EPBs outside the ~115°E region were not observed.
In order to provide much more detailed comparison between the all-sky imager and
TEC measurements, we chose those TEC variations of corresponding geographical area
and time of each airglow imaging of Figure 3 in Figure 5. In Figure 5, the TEC
variations show that the EPBs at ~105° E appeared near 05:30 LT, which correspond to
EPB "b1" and "b2" observed by the all-sky imager. In Figure 5, TEC depletions move
away from the 106° E with time (The black vertical line represents the 106°E in Figure
5), which is consistent with the movement of EPBs observed by the airglow imager.
Meanwhile, the northernmost part of the depletion of ~105°E expanded to ~25°N at
06:20 LT (The black horizontal line represents the 25°N in Figure 5), which also agreed
well with the observations of the all-sky imager. Interestingly, TEC variations show that
the northernmost of EPBs of ~105°E extended beyond 25°N after 06:20 LT. We can see
that the northernmost of them reached about 28°N at 07:10 LT in Figure 4. In other
words, TEC variations show that the depletions of ~105°E were still there after 06:21
LT, and kept developing after sunrise, but vanished after ~08:00 LT. These
observational results shown that the life time of those EPBs exceeds 3 hours.

**4.  Discussion**
In this study we showed an special event of EPBs which was simultaneously observed
by the all-sky imager and the ground GNSS network in the south China. One interesting



feature is that the EPBs started to appear near sunrise hours. Afterward, they kept
developing until they totally vanished. During their life time, the EPBs moved from
east to west. Those EPBs occurred in the recovery phase of the geomagnetic storm,
which indicates that the prompt penetration electric fields (PPEF) and disturbance
dynamo (DDEF), as well as disturbed neutral wind circulation may play an import role
in triggering the EPBs.
The drift velocities of EPBs were shown in Figure 6. We used the cross sections
(keogram) (Figures 6 (a), (c), and (e)) of the airglow images to separately calculate
meridian velocities (Figure 6(b)) of "b1" and zonal velocities of "b1" at ~ 22°N (Figure
6(d)) and ~19°N (Figure 6(f)) geographical latitudes. Figure 6(a) illustrates the N-S
cross sections (between 104°E and 105°E) of the airglow images shown in Figure 3.
Figure 6(c) illustrates the W-E cross sections (between 21.5°N and 22°N) of the airglow
images, and Figure 6(e) illustrates the W-E cross sections (between 18.5°N and 19°N).
We separately calculated poleward and zonal velocities of "b1" based on the position
of it changed over time in Figure 6(a), Figure 6(c) and Figure 6(e). The initial poleward
and zonal velocities of "b1" were about 200 m/s and 60 m/s, respectively. Horizontal
drift of EPB is also an important issue, which is often related to the background zonal
plasma drift (Fejer et al., 2005; Eccles, 1998). The westward motion of the F-region
should be caused by the ionospheric dynamo process in the early morning (Kil et al.,
2000; Sheehan and Valladares, 2004). The drift direction of background zonal plasma
drift has a reversal (eastward to westward) near dawn (Fejer et al., 2005). In our case,
all EPBs emerged after 05:00 LT. The background plasma should drift westward during
the early morning hours. So, it could partly explain why the observed EPBs drifted
westward. In addition, the disturbed westward neutral winds can also contribute to the
westward drifting of EPBs. Xiong et al. (2015) found that the disturbance winds were
mainly towards westward at low latitudes, most prominent during early morning hours.
Abdu et al. (2003) found that the westward drift of an EPB was most likely caused by
westward zonal winds during a geomagnetic storm. Makela et al. (2006) found that the
eastern wall of EPBs can become unstable due to the westward and equatorward neutral



winds associate with wind surges. In Figure 3, a sub-branch of dark bands first occurred
at the eastern wall of "b1", indicated secondary instabilities developed at the eastern
edge, most likely due to the westward disturbance winds.
In Figure 7, we used the Thermosphere-Ionosphere-Electrodynamics General
Circulation Model (TIE-GCM) to simulate the horizontal winds on 08 November 2015
under magnetically active conditions, and the latitude versus longitude distribution of
zonal wind velocities are shown at different times. The winds at 250 km are shown, and
the spatial coverage has been confined to 0° - 40° N latitude and 90° - 120° E longitude.
The dashed rectangles represent the location of "b1" and "b2" at different times. In
Figure 7, we can see that the horizontal winds at low latitudes are mainly westward,
which is consistent with the motion of EPBs in this case. As already discussed above,
the westward drift of those EPBs is possibly caused by the westward disturbance winds.
Besides, the zonal winds computed from TIE-GCM shown in Figure 7 are smaller than
the zonal drifts of EPBs shown in Figure 6. This is because zonal drift value of EPBs
was controlled by background zonal winds and ionospheric electric field (Haerendel et
al., 1992; Eccles, 1998). The value differences between simulation and zonal drifts of
EPBs should be influenced by ionospheric electric field.
As reported, most of the EPBs start to occur at pre-midnight hours. There were a very
limited number of studies that used data from radar or satellite to report the occurrence
of EPB close to sunrise hours (e.g., Fukao et al., 2003; Huang et al., 2013; Zhou et al.,
2016). However, until now, there has been no observation result of EPBs around sunrise
using optical remote sensing. In fact, it is very difficult to observe EPB near sunrise by
an all-sky imager. Often, EPBs start to develop shortly after sunset and vanish before
sunrise. Even though some EPBs occur around sunrise in their initial stage, they
disappear when they drift eastward into the daytime. And almost no report shows that
the EPBs still kept developing after sunrise. In our case, the developing EPB was first
observed at about 05:30 LT (near dawn) by both the all-sky imager and the GNSS
network. Our observational results show that they kept developing after sunrise, and
vanished about one hour after sunrise. Those EPBs should be occurred near sunrise,



which is different from post-sunset EPBs. Their development stages lasted for at least
about 3 hours.
In the rest, we try to explain why the EPBs occurred near sunrise. During the storm
time, disturbance winds can affect the low-latitude ionospheric electrodynamics as well
as the zonal drift of an EPB. The DDEF will drive plasma drift to move upward at
nighttime during the development phase of storm (Blanc and Richmond, 1980).
Meanwhile, a number of studies found the that high latitude electric fields can penetrate
into the middle and low-latitude ionosphere as PPEF when IMF $Bz$ turns southward or
northward (Kelley et al., 1979; Scherliess and Fejer, 1997; Cherniak and Zakharenkova,
2016; Carter et al., 2016; Patra et al., 2016; Katamzi-Joseph et al, 2017). For the storm
event, after IMF $Bz$ turned southward at ~12:00 LT 07 November 2015, there was long
duration and high AE in storm time. A DDEF should be present at recovery phase of
storm time. And it reversed ambient electric field from westward to eastward near
sunrise, which enhanced height of bottomside of the ionosphere $F$-region. Meanwhile,
the northward turning of IMF $Bz$ at ~04:00 LT 08 November 2015 caused over-
shielding electric field, which produced an eastward PPEF into the low-middle latitude
ionosphere. The eastward electric field also moved the $F$ region ionosphere to higher
altitudes via vertical $E{\times}B$ drifts. In Figure 2(e), the increased height of bottomside of
the ionosphere $F$-region can be seen at Fuke. In low latitude region, one of the necessary
conditions for the generation of EPBs is that the $F$ layer should be uplifted to a higher
altitude, where the RTI becomes unstable and forms EPBs. The $F$ layer height is largely
determined by the eastward field via the vertical $E{\times}B$ drift (Dabas et al., 2003).
In this study, EPBs were initially observed by the all-sky imager at about 05:15 LT. We
think that only a portion of the EPBs were observed in our study, as EPB usually extend
along the whole magnetic flux-tube. It also means that the EPBs should possibly occur
before 05:15 LT at equatorial latitude. But due to the lack of observations at equator,
we cannot provide direct evidence about their generation. However, as shown in our
Figure 8, we also found that spread $F$ began to appear in the ionograms from the
digisonde at Fuke after 05:15 LT, which indicates that those EPBs occurred in the region



of southeastern Qujing (Note that Fuke is to the southeast of Qujing). Bottomside of
the ionospheric F-region at Fuke was rapidly elevated from ~250 km to ~290 km near
sunrise on 08 November 2015. The rapidly elevated height of the ionosphere can cause
stronger RTI at the bottom of the ionosphere F-region, which is beneficial to the
formation of EPB. The initial occurring time of EPBs of this case should be during this
time. Unfortunately, we do not have more observations in the southeast of Fuke. We
used the TIE-GCM to simulate the height of hmF2 at lower latitude on 08 November
2015. Figure 9 shows the hmF2 as a function of longitude and latitude at different times.
The model results plotted are in a geographic range from 0° to 40° N in latitude and
from 90° to 120° E in longitude. In Figure 9, we can see that hmF2 southeast of (the
dashed rectangles) Qujing was rapidly elevated to higher altitudes near sunrise. In other
words, when the IMF $Bz$ turned northward at about 04:00 LT, the ionosphere in some
regions southeast of Qujing could be rapidly elevated to higher altitudes at this time.
Those EPBs occurred in the same time period as highlighted by the green rectangular
area in Figure 2. Previous studies have reported that the occurrence of the dawn
enhancement in the equatorial ionospheric vertical plasma drift (Zhang et al., 2015,
2016). They found that the enhancement of the ionospheric vertical plasma drift occurs
around dawn. They suggested that the vertical plasma drifts can be enhanced near
sunrise in a way similar to the PRE near sunset. Fejer et al. (2008) found that the
nighttime disturbance dynamo drifts are upward, and have the largest values near rise.
In our case, the model simulations and observations both show an increasing of the
height of the ionosphere around sunrise. The enhancement of low-latitude ionospheric
vertical plasma drift caused by DDEF and PPEF associated with the geomagnetic storm
should play a vital role in triggering those EPBs. Our results also provide evidence of
the enhancement of low-latitude ionospheric vertical plasma drift around sunrise, which
should be the main reason of the EPBs generation near dawn.
In addition, some interesting features of EPBs are also shown in Figure 3 in that the
EPBs showed also break and recombination processes. In Figure 6(f), at latitude of
19°N, the zonal velocity of "b1" was about 60-70 m/s between 05:20 LT and 06:15 LT.



However, at the latitude of 22°N (Figure 6(d)), the zonal velocity of "b1" was decreased
from about 70 m/s to about 50 m/s between 05:20 LT and 05:45 LT. After 05:45 LT, its
velocity began to increase from ~50 m/s to ~70 m/s from 05:45 LT to 06:00 LT. Then,
it kept a velocity of ~70 m/s. Owning to the fact that the zonal velocity at higher
latitudes was smaller than that at low latitudes before 05:45 LT, "b1" had a break
process of EPBs during this period. After 05:45 LT, the zonal velocity at higher latitude
was bigger than that at lower latitude, "b1" exhibited a recombination process of EPBs
after 06:03 LT. The above results indicate that the break and recombination processes
of EPBs should be caused by the different drift velocities of the background plasma at
different latitudes.

**5. Summary**
In this paper, a special EPB event was observed by an all-sky imager and the GNSS
network in the southern China. The evolution processes and characteristics of those
EPBs were studied in detail. Our main findings are summarized as below:
(1) The observed EPBs on 08 November 2015 emerged before sunrise and kept

developing. They dissipated at about one hour after sunrise (~ after 08:00 LT) and

the development stage lasted for at least about 3 hours. The evolution of EPBs

developing around sunrise was observed for the first time by an all-sky imager and

the GNSS network.

(2) They occurred in the recovery phase of a geomagnetic storm. The enhancement of

background ionospheric vertical plasma drift was also observed near sunrise. The

rapid uplift of the ionospheric caused by the geomagnetic storm should be the main

reason for triggering the EPBs.

(3) During the development, the EPBs drifted westward rather than eastward, The TIE-

GCM simulation suggested that the westward drift of EPB is related to the westward

disturbance winds.

(4) The EPB exhibited also break and recombination processes during its development.



**Acknowledgement**
This work was supported by the Open Research Project of Large Research
Infrastructures of CAS – "Study on the interaction between low/mid-latitude
atmosphere and ionosphere based on the Chinese Meridian Project" and the Chinese
Meridian Project, and the National Natural Science Foundation of China (41674152
and 41331069). The airglow and digisonde data were downloaded from
http://data.meridianproject.ac.cn/. The airglow data used in this study can be obtained
by contacting the corresponding author. We acknowledge the use of GNSS data from
the Crustal Movement Observation Network of China (CMONOC, http:// neiscn.org/)
and the data could be obtained upon request. We thank H. Liu from Macao University
of Science and Technology for processing the GNSS data. The IMF, AE, AL, AU, KP,
and SYM/H data are obtained from the CDAWeb (https://cdaweb.sci.gsfc.nasa.gov/)
and the WDC for geomagnetism at Kyoto University (https://wdc.kugi.kyoto-u.ac.jp/).
The National Center for Atmospheric Research is sponsored by the National Science
Foundation.



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



**Figure Captions**
**Figure 1.** The location of observation instruments. The red star denotes the geographic
location of the all-sky imager at Qujing (25° N, 104° E). The blue circle denotes the
field of view of the all-sky imager at an altitude of 250 km. The green dot denotes the
geographic location of the digisond at Fuke (19.5° N, 109.1° E). The red dotted line
represents the magnetic equator.

**Figure 2.** (a) $Kp$ indexes, (b) the interplanetary magnetic field (IMF) $Bz$, (c) SYM/H,
and (d) AE, AU, AL during 06-08 November 2015. (e) The variations of h'F obtained
from the digisond at Fuke on 06-08 November 2015.

**Figure 3.** Images of equatorial plasma bubbles from the Qujing site between 05:15 LT
and 06:21 LT on 08 November 2015. The observed images were mapped into
geographical coordinates by assuming that the airglow emission layer was at an altitude
of ~250 km. The white vertical line is a reference line of 106° E and horizontal line is
a reference line of 25° N.

**Figure 4.** Total electron content residuals over China and adjacent areas with 10 minute
interval during 04:30 – 08:20 LT on 08 November 2015. The black horizontal line is a
reference line of 25° N.

**Figure 5.** Total electron content residuals correspond to each image of Figure 3. The
black horizontal line is a reference line of 25° N. The black vertical line is a reference
line of 106° E.

**Figure 6.** (a) N-S cross sections (between 104°E and 105°E) of the airglow images on
08 November 2015. (c) W-E cross sections (between 21.5°N and 22°N) of the airglow
images. (e) W-E cross sections (between 18.5°N and 19°N) of the airglow images. (b)
The variations of the meridian velocities of "b1" with local time. (d) and (f) The



variations of the zonal velocities of "b1" at ~ 22°N and ~19°N geographical latitudes,
respectively.

**Figure 7.** Contours of nighttime zonal winds at 250 km in a range from 0° to 40° N in
latitude and from 90° to 120° E in longitude during 08 November 2015. The dashed
rectangles represent the location of EPBs.

**Figure 8.** The ionograms observed by the digisonde at Fuke between 04:00 LT and
07:30 LT on 08 November 2015.

**Figure 9.** The height of hmF2 in a range from 0° to 40° N in latitude and from 90° to
120° E in longitude during 08 November 2015. The red star represent the location of
all-sky imager. The dashed rectangles represent the region of southeastern Qujing.



Figure 1

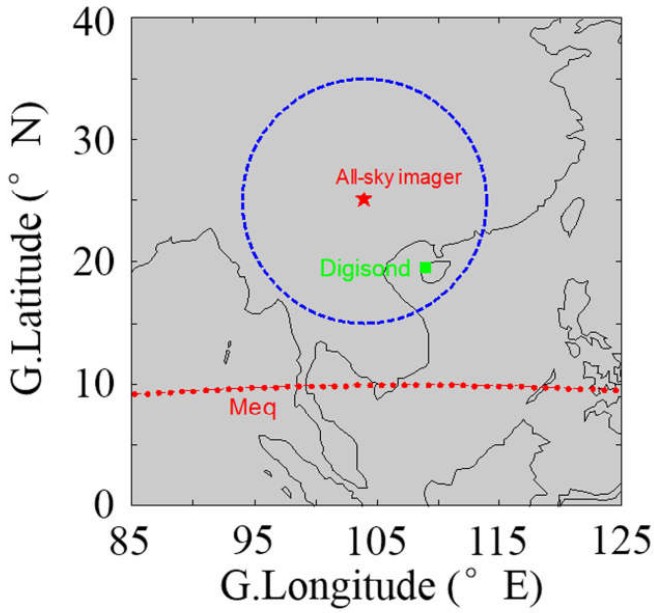



Figure 2

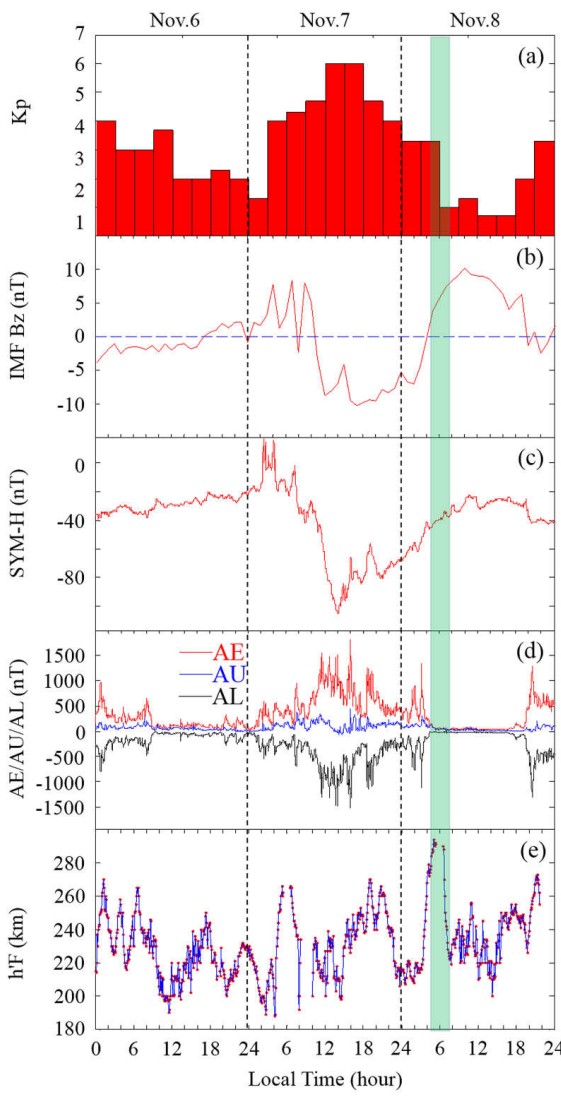



Figure 3

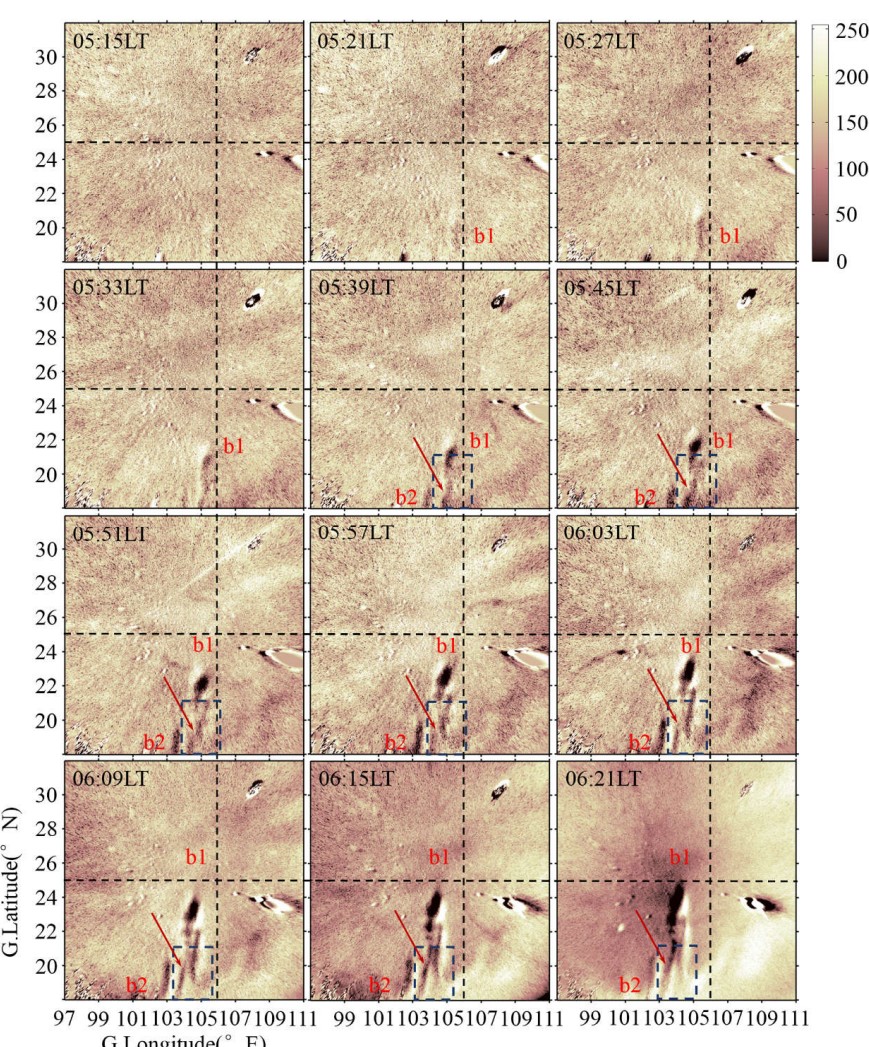





Figure 4

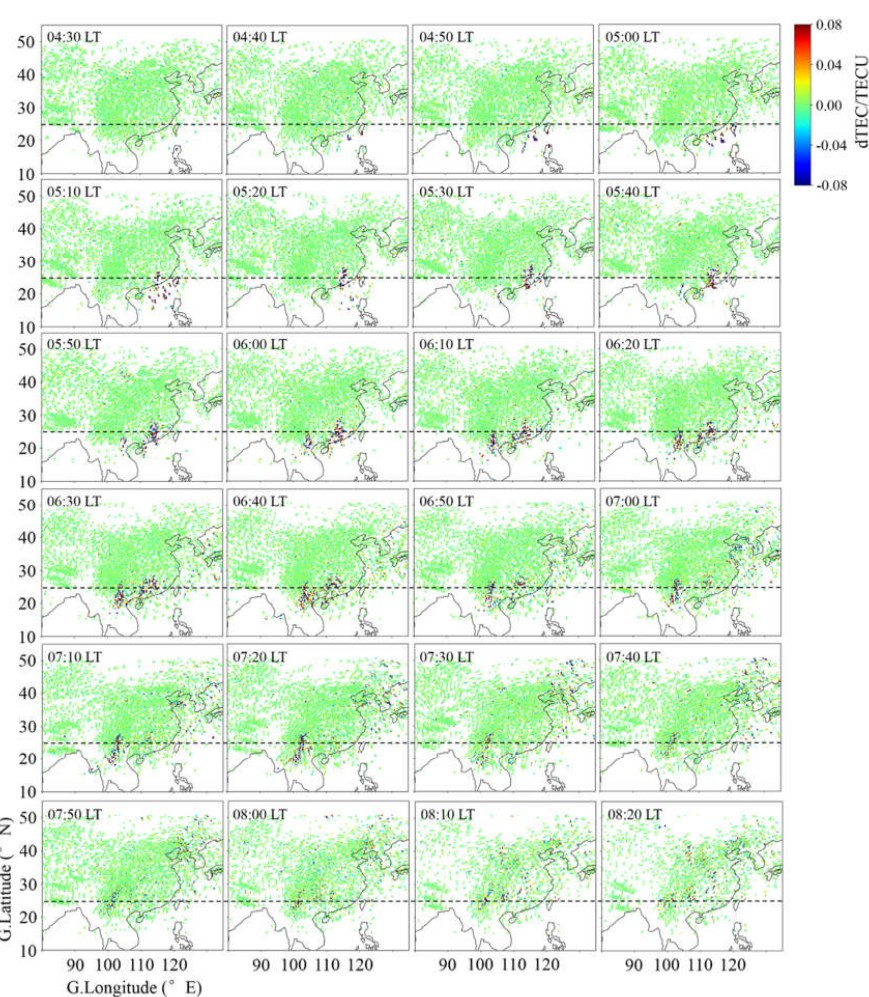



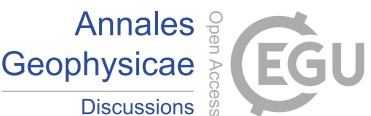

Figure 5

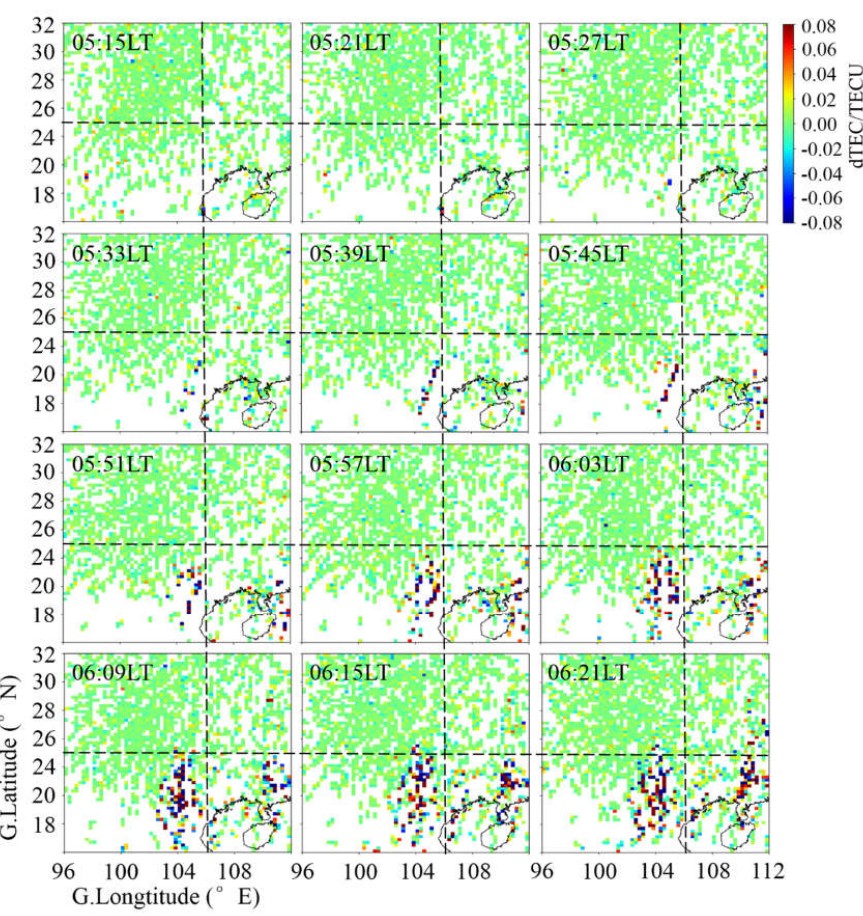



Figure 6

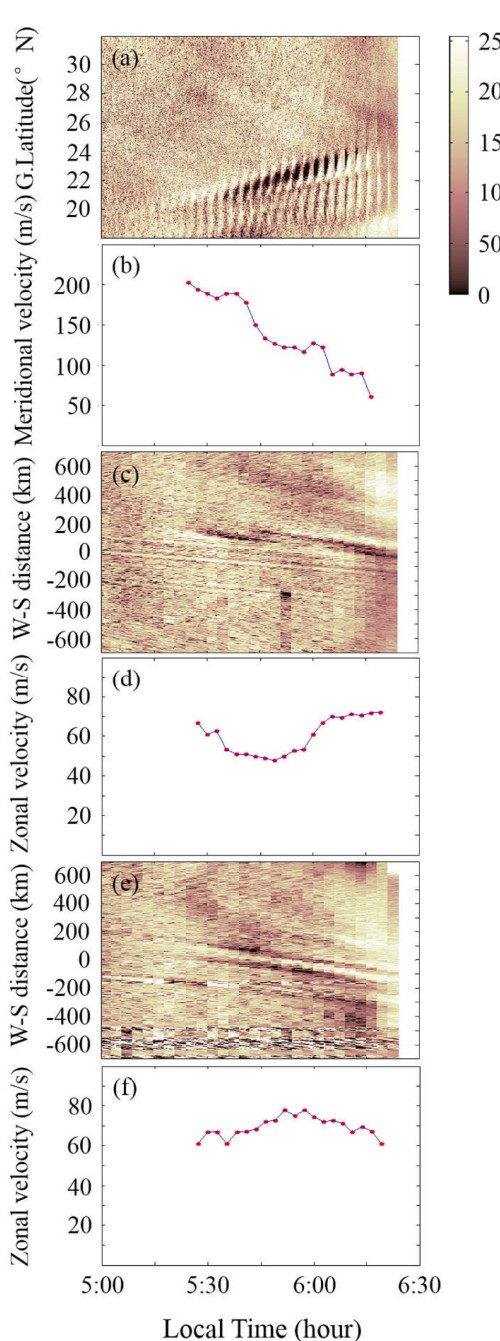

Local Time (hour)




Figure 7

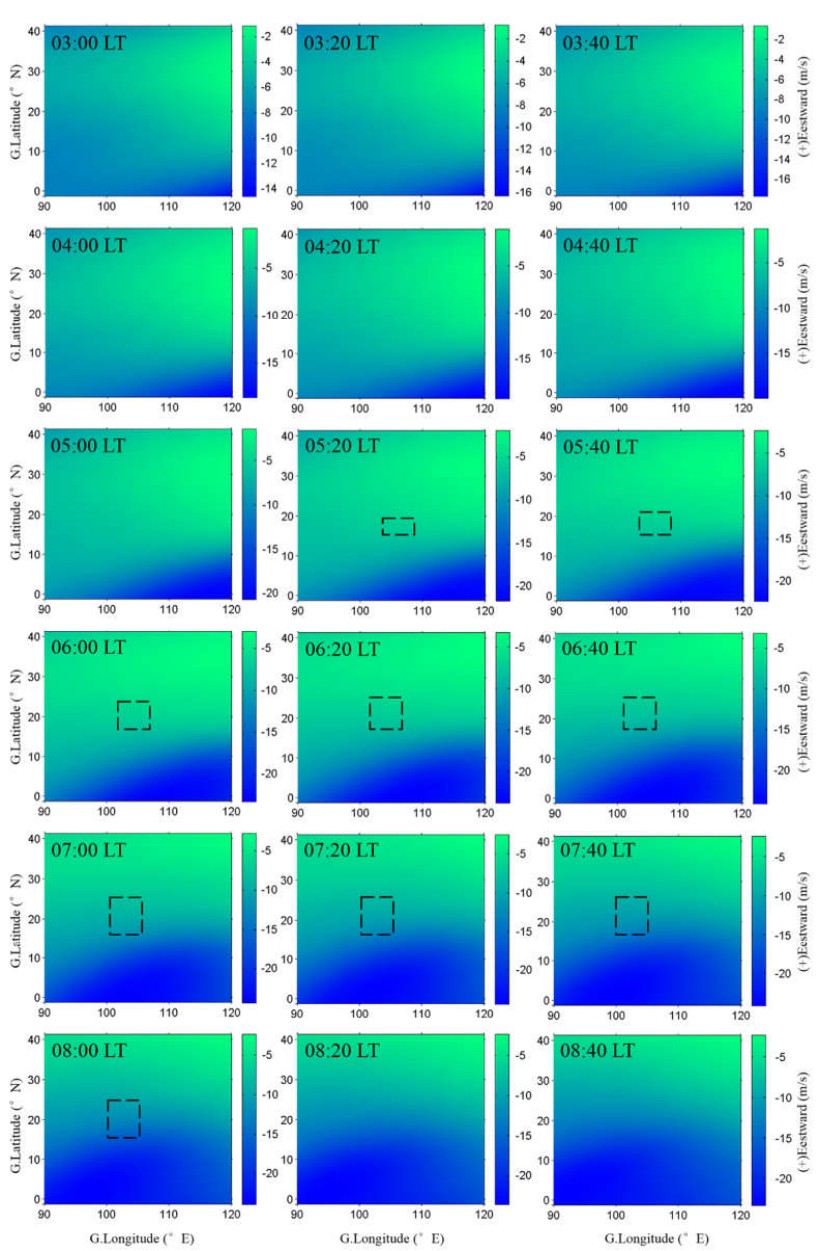



Figure 8

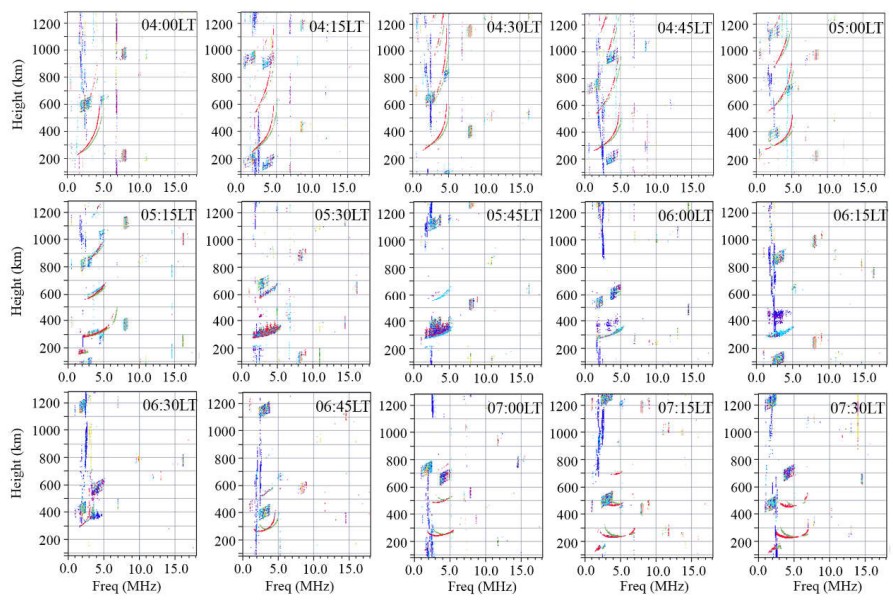



Figure 9

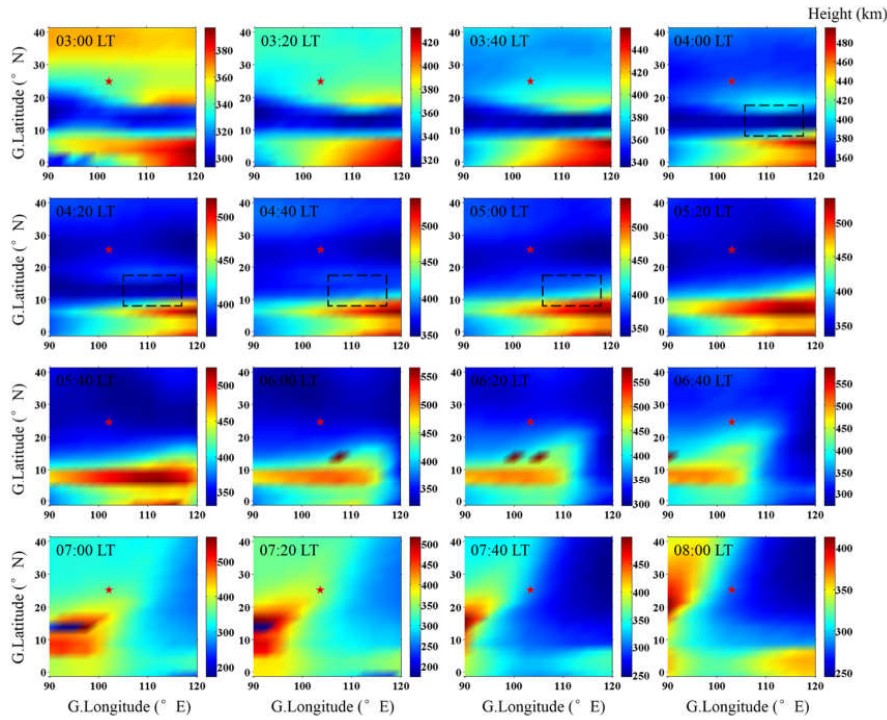