# Peer review of "Equatorial Plasma Bubbles Developing Around Sunrise Observed by an All-Sky Imager and GNSS Network during the Storm Time"

_Annales Geophysicae, 2019_

## Short Comment (SC1) · 27 Aug 2019

This paper reports all-sky airglow and GNSS-TEC observations of plasma bubbles growing around sunrise terminator during a magnetic storm. This work could contribute to study of effects of magnetic storm on ionospheric disturbances. Therefore, this paper is worth publishing in this journal. However, the followings need to be addressed before its publication.

Specific comments: – "recombination": This reviewer recommends the authors to use a term "merging". "Recombination" is confusing because "recombination" is widely used to represent reaction of ions with electrons resulting in neutralization. "Merging"

[Figure]

is used commonly compared to "recombination". See the following references:

Narayanan, V. L., S. Gurubaran, and K. Shiokawa (2016), Direct observational evidence for the merging of equatorial plasma bubbles, J. Geophys. Res. Space Physics, 121, 7923–7931, doi:10.1002/2016JA022861.

Huba, J. D., T.-W. Wu, and J. J. Makela (2015), Electrostatic reconnection in the ionosphere, Geophys. Res. Lett., 42, 1626–1631, doi:10.1002/2015GL063187.

Huang, C.-S., J. M. Retterer, O. de La Beaujardiere, P. A. Roddy, D. E. Hunton, J. O. Ballenthin, and R. F. Pfaff (2012), Observations and simulations of formation of broad plasma depletions through merging process, J. Geophys. Res., 117, A02314, doi:10.1029/2011JA017084.

- l. 55, "and the background ionospheric/thermosphere": Describe concretely which parameter the authors mean. Does the authors mean vertical gradient of plasma density at the bottomside of the F region or ion-neutral collision frequency?

- l. 96, "Hall electric field": It is better to add more detailed explanation of the Hall electric field.

– ll. 110-111, Figure 1: Field-of-view (FOV) is shown by a circle in Figure 1. It would be better to describe the zenith angle corresponding to the circle shown as FOV.

– l. 122: Describe minimum and maximum frequency (or period) of the band-pass filter.

– ll. 176-197: The authors describe that TEC depletion can be seen in Figures 4 and 5. However, TEC variations in these figures show positive and negative values rather than depletion. Spatial scale of the TEC variations seen in the figures is small. Therefore, the TEC variation corresponds to the plasma density irregularities existing within plasma bubbles. If the authors show ROTI (Rate of TEC change Index), structure of the plasma bubbles can be seen clearly as ROTI enhancements. See the following paper.

Buhari, S. M., Abdullah, M., Hasbi, A. M., Otsuka, Y., Yokoyama, T., Nishioka, M., and Tsugawa, T. (2015), Continuous generation and two‑dimensional structure of equatorial plasmabubbles observed by high‑density GPS receivers in Southeast Asia, J. Geophys. Res. Space Physics, 119, pages 10,569– 10,580. doi:10.1002/2014JA020433.

– ll. 229-232: Explain a reason why the eastern wall of EPB is unstable when the wind blows westward and equatorward. When the wind blow westward, and thus the wind-induced Pedersen current flows downward, gradient-drift instability can occur at the eastern wall of EPB, where the plasma density gradient is eastward. On the other hand, how does the equator ward wind work?

– ll. 233-247, "This is because zonal drift value of EPBs ... EPBs should be influenced by ionospheric electric field.": The authors point out that the drift velocity of EPB is smaller than the wind, and argue the reason of this difference. However, this reviewer cannot understand what the authors are describing. If the F-region dynamo process completely works, the ExB drift velocity is equal to the wind velocity. Does the authors mean that electric field generation through the F-region dynamo is not completed and thus the ExB drift is smaller than the wind velocity? Otherwise, does the authors consider another electric field, which is different from the dynamo electric filed induced by the wind?

– ll. 255-256: The authors point out the EPBs kept developing after sunrise. Generally, it is considered that after sunrise, the photoionization due to the Solar EUV radiation produce the plasma in the ionosphere and fill the plasma depletion of EPB. In order to compare the time of sunrise, it would be worth showing local time variation of the absolute TEC, to compare the time of EPB existence with the time of rapid TEC increase at sunrise.

– l. 265, "during the development phase of storm": What is the development phase of storm? Is it "main phase of magnetic storm"? Why does the DDEF appear only during

the development phase of storm?

Minor comments: – l. 127: "Digisond" –> "Digisonde" – l. 308: "rise" may be "sunrise". – Figure 6: Legend of vertical axis in Figures 6c and 6e is "W-S distance". It should be "W-E distance". Furthermore, describe positive eastward.

---

## Referee Comment (RC1) · Anonymous Referee #1 · 16 Dec 2019

Referee's report on

"Equatorial Plasma Bubbles Developing Around Sunrise Observed by an All-Sky Imager and GNSS Network during the Storm Time"

by Kun Wu et al.

This paper reports all-sky airglow and GNSS-TEC observations of plasma bubbles forming around the sunrise terminator during the recovery phase of a magnetic storm. This represents a contribution to the study of magnetic-storm effects on ionospheric disturbances. Therefore, this paper is worth publishing in the journal following minor revision as described below:

[Figure]

'Break' and 'recombination' are nonstandard terminology for the phenomena the authors describe; for 'break' people usually say 'bifurcation', and for 'recombination', 'merging'. Actually, the 'break' that appears in the all-sky images looks to me like it could be the development of another bubble, or possibly the emergence of an arm on the side of their main bubble. These phenomena were discussed in detail, with simulations, by Huang et al. ( Huang, C.-S., J. M. Retterer, O. de La Beaujardiere, P. A. Roddy, D. E. Hunton, J. O. Ballenthin, and R. F. Pfaff (2012), Observations and simulations of formation of broad plasma depletions through merging process, J. Geophys. Res., 117, A02314, doi:10.1029/2011JA017084.)

The variation of the zonal drift within plasma bubbles with both solar activity and geomagnetic variations was discussed by Huang and Roddy (Huang, C.-S., and P. A. Roddy (2016), Effects of solar and geomagnetic activities on the zonal drift of equatorial plasma bubbles, J. Geophys. Res. Space Physics, 121, 628–637, doi:10.1002/2015JA021900.), which would be a useful reference here.

Finally, the presence of bubbles around sunrise was investigated thoroughly in the in-situ observations of the plasma density by the C/NOFS satellite, and those studies should be referenced here: Huang, C.-S., O. de La Beaujardiere, P. A. Roddy, D. E. Hunton, J. O. Ballenthin, and M. R. Hairston (2013),Long-lasting daytime equatorial plasma bubbles observed by the C/NOFS satellite,J. Geophys. Res. Space Physics,118,2398–2408, doi:10.1002/jgra.50252.

---

## Editor Comment (EC1) · Keisuke Hosokawa (Editor) · 20 Dec 2019

The short comment from Dr. Otsuka should regarded as a referee comment. Thus, now we have two reviews for this manuscript. The interactive discussion will be closed and editor's decision will be made as a next step.

---

## Author Comment (AC3) · 23 Dec 2019

Dear Editor:

Thanks for your reminder. Since we are not familiar with the manuscript review process, the reply to the reviewer was delayed. We are so sorry for that.

Yours sincerely,

Authors

---

## Author Response (AR1)

**Reply to Yuichi Otsuka and Referee #1**

**(1) : Reply to Yuichi Otsuka**

This paper reports all-sky airglow and GNSS-TEC observations of plasma bubbles growing around sunrise terminator during a magnetic storm. This work could contribute to study of effects of magnetic storm on ionospheric disturbances. Therefore, this paper is worth publishing in this journal. However, the followings need to be addressed before its publication.

Reply: Thank you for your positive comments. All the comments from you have been considered in the revised manuscript. And with the corrections made, we hope it's accepted for publication in Annales Geophysicae now.

Specific comments: – "recombination": This reviewer recommends the authors to use a term "merging". "Recombination" is confusing because "recombination" is widely used to represent reaction of ions with electrons resulting in neutralization. "Merging" is used commonly compared to "recombination". See the following references:

Narayanan, V. L., S. Gurubaran, and K. Shiokawa (2016), Direct observational evidence for the merging of equatorial plasma bubbles, J. Geophys. Res. Space Physics,121, 7923–7931, doi:10.1002/2016JA022861.

Huba, J. D., T.-W. Wu, and J. J. Makela (2015), Electrostatic reconnection in the ionosphere, Geophys. Res. Lett., 42, 1626–1631, doi:10.1002/2015GL063187.

Huang, C.-S., J. M. Retterer, O. de La Beaujardiere, P. A. Roddy, D. E. Hunton, J. O. Ballenthin, and R. F. Pfaff (2012), Observations and simulations of formation of broad plasma depletions through merging process, J. Geophys. Res., 117, A02314, doi:10.1029/2011JA017084.- l.

Reply**:** Thanks for this suggestion. After reading above references, we used "merging" to replace "recombination" and cited those references in the revised manuscript.

**- l. 55, "and the background ionospheric/thermosphere": Describe concretely which parameter the authors mean. Does the authors mean vertical gradient of plasma density at the bottomside of the F region or ion-neutral collision frequency?**

Reply: We want to address that the growth rate of Rayleigh-Taylor instability (RTI) will be influenced by the vertical gradient of plasma density at the bottomside of F region, and also the change of ion-neutral collision frequency. We have revised related texts at lines 55-56.

**- l. 96, "Hall electric field": It is better to add more detailed explanation of the Hall electric field.**

Reply: In the revised manuscript, we added related explanation of "Hall electric field" at lines 95-99.
"Santos et al. (2016) also showed some EPBs of zonal drifts reversal (eastward to westward) during a geomagnetic storm, and they suggested the reversal was caused by a vertical Hall electric field which induced by a zonal prompt penetration electric field (PPEF) in the presence of enhanced conductivity in the E region during night."

Santos, A. M., Abdu, M. A., Souza, J. R., Sobral, J. H. A., Batista, I. S., and Denardini, C. M.: Storm time equatorial plasma bubble zonal drift reversal due to disturbance Hall electric field over the Brazilian region, J. Geophys. Res., 121, 5594–5612, https://doi.org/10.1002/2015JA022179, 2016.

**– ll. 110-111, Figure 1: Field-of-view (FOV) is shown by a circle in Figure 1. It would be better to describe the zenith angle corresponding to the circle shown as**

**FOV.**

Reply: In Figure 1, the blue circle represents the projected regions with a radius of ~900 km [about 140° field of view (FOV)] of the all-sky imager at an altitude of 250 km. We have revised related texts at lines 113-115.

**– l. 122: Describe minimum and maximum frequency (or period) of the band-pass filter.**

Reply: The minimum and maximum period of the band-pass filters we used are 2 min and 12 min, respectively. In the revised manuscript, we added related content at lines 126-127.

**– ll. 176-197: The authors describe that TEC depletion can be seen in Figures 4 and 5. However, TEC variations in these figures show positive and negative values rather than depletion. Spatial scale of the TEC variations seen in the figures is small. Therefore, the TEC variation corresponds to the plasma density irregularities existing within plasma bubbles. If the authors show ROTI (Rate of TEC change Index), structure of the plasma bubbles can be seen clearly as ROTI enhancements. See the following paper.**
**Buhari, S. M., Abdullah, M., Hasbi, A. M., Otsuka, Y., Yokoyama, T., Nishioka, M., and Tsugawa, T. (2015), Continuous generation and twoâA˘ Rdimensional structure of equatorial plasmabubbles observed by highâA˘ Rdensity GPS receivers in Southeast Asia, J. Geophys. Res. Space Physics, 119, pages 10,569– 10,580.doi:10.1002/2014JA020433.**

Reply: Thanks for your advice. We calculated the ROTI variations (Figure 6) which correspond geographical area and time of each airglow imaging in the revised manuscript. In the Figure 6, we can clearly ROTI enhancement from structure of the EPBs.

[Figure]

**Figure 6.** Two-dimensional map of rate of TEC index (ROTI) correspond to each image of Figure 3. The black horizontal line is a reference line of 25° N. The black vertical line is a reference line of 106° E.

– ll. 229-232: **Explain a reason why the eastern wall of EPB is unstable when the wind blows westward and equatorward. When the wind blow westward, and thus the wind-induced Pedersen current flows downward, gradient-drift instability can occur at the eastern wall of EPB, where the plasma density gradient is eastward. On the other hand, how does the equator ward wind work?**

Reply: Due to Coriolis force, the enhanced equatorward wind at disturbed periods will have also a westward component, which will work on the eastward wall of EPB, causing the secondary instabilities. Similar finding of secondary instability happened at the eastward wall of EPB has been earlier reported by Makela et al. (2006), by using airglow imagers. In the revised manuscript, we added related context at lines 251-256.

Makela, J. J., Kelley, M. C., and Nicolls, M. J.: Optical observations of the development of secondary instabilities on the eastern wall of an equatorial plasma bubble J. Geophys. Res., 111, A9, https://doi.org/10.1029/2006JA011646, 2006.

**– ll. 233-247, "This is because zonal drift value of EPBs ... EPBs should be influenced by ionospheric electric field.": The authors point out that the drift velocity of EPB is smaller than the wind, and argue the reason of this difference. However, this reviewer cannot understand what the authors are describing. If the F-region dynamo process completely works, the ExB drift velocity is equal to the wind velocity. Does the authors mean that electric field generation through the F-region dynamo is not completed and thus the ExB drift is smaller than the wind velocity? Otherwise, does the authors consider another electric field, which is different from the dynamo electric filed induced by the wind?**

Reply: Here, our understanding is that the zonal plasma drifts are affected by the vertical electric fields generated by the $E$ and $F$ region wind dynamo (Haerendel et al., 1992). The $E$ and $F$ region dynamo effects can be examined by using a simplified formula from Eccles et al. (1998): $V_\varphi = U_\varphi^P = \frac{\Sigma_P^F U_\varphi^{PF} + \Sigma_P^E U_\varphi^{PE}}{\Sigma_P}$. Where $V_\varphi$ is the zonal plasma drift speed, $U_\varphi^P$ is the Pedersen conductivity-weighted neutral zonal winds, $\Sigma$ is the field-line-integrated total ionospheric conductivity. $E$ and $F$ refer to the $E$ and $F$ region, respectively. $P$ represents the Pedersen component. During nighttime, the $E$ layer is quickly recombined and $F$ layer dynamo plays a dominant role. So, the zonal drift value of EPBs should mainly be related to $\frac{\Sigma_P^F U_\varphi^{PF}}{\Sigma_P}$. The simulation of Figure 7 reflect $U_\varphi^{PF}$. The difference between the model simulated background zonal winds and the derived zonal drifts of EPBs from airglow images is possibly due to that the model simulation provide mainly reflect a general trend of the wind, but not the exact wind velocity in reality.

Eccles., V. J.: A simple model of low-latitude electric fields, J. Geophys. Res., 103, 26699-26708, https://doi.org/10.1029/98JA02657, 1998.

Haerendel, G., Eccles, J. V., and Çakir, S.: Theory for modeling the equatorial evening ionosphere and the origin of the shear in the horizontal plasma flow, J. Geophys. Res., 97, 1209–1223, https://doi.org/10.1029/91JA02226, 1992.

**– ll. 255-256: The authors point out the EPBs kept developing after sunrise. Generally, it is considered that after sunrise, the photoionization due to the Solar EUV radiation produce the plasma in the ionosphere and fill the plasma depletion of EPB. In order to compare the time of sunrise, it would be worth showing local time variation of the absolute TEC, to compare the time of EPB existence with the time of rapid TEC increase at sunrise.**

Reply: In the revised manuscript, we added it in Figure 5. The TEC depletions showed that EPBs existed after sunrise and they disappeared after 07:45 LT. These results showed that they vanished about one hour after sunrise. Their life time lasted for at least about 3 hours.

**– l. 265, "during the development phase of storm": What is the development phase of storm? Is it "main phase of magnetic storm"? Why does the DDEF appear only during the development phase of storm?**

Reply: We are sorry for the misleading description in our previous manuscript. Once DDEF established, the DDEF could be last from hours to couple of days (Richmond et al., 2003). So, we rewrote the sentence in the revised manuscript. It is modified for "The DDEF caused by storm will drive plasma drift to move upward during nighttime (Blanc and Richmond, 1980)".

Blanc, M., and Richmond, A. D.: The ionospheric disturbance dynamo, J. Geophys. Res., 85, A4, https://doi.org/10.1029/JA085iA04p01669, 1980.

Richmond, A. D., Peymirat, C., and Roble, R. G.: Long‑lasting disturbances in the equatorial ionospheric electric field simulated with a coupled magnetosphere‑ionosphere‑thermosphere model, J. Geophys. Res., 108, A3, https://doi.org/10.1029/2002JA009758, 2003.

**Minor comments:**

**-l. 127: "Digisond" –> "Digisonde"**

**-l. 308: "rise" may be "sunrise".**

**Figure 6: Legend of vertical axis in Figures 6c and 6e is "W-S distance". It should be "W-E distance". Furthermore, describe positive eastward.**

Reply: Thank you for these detailed suggestions. We used "Digisonde" to replace "Digisond" and used "sunrise" to replace "rise". The previous Figure 6 has been updated as Figure 7 in the revision.

[Figure]

**Figure 7.** (a) N-S cross sections (between 104°E and 105°E) of the airglow images on 08 November 2015. (c) W-E cross sections (between 21.5°N and 22°N) of the airglow images. (e) W-E cross sections (between 18.5°N and 19°N) of the airglow images. (b) The variations of the meridian velocities of "b1" with local time. (d) and (f) The variations of the zonal velocities of "b1" at ~ 22°N and ~19°N geographical latitudes, respectively.

**(2): Reply to Referee #1**

**This paper reports all-sky airglow and GNSS-TEC observations of plasma bubbles forming around the sunrise terminator during the recovery phase of a magnetic storm. This represents a contribution to the study of magnetic-storm effects on ionospheric disturbances. Therefore, this paper is worth publishing in the journal following minor revision as described below:**

Reply: We are grateful to the reviewer for useful comments on our manuscript. All the comments from you and also the other reviewer have been considered in the revised manuscript. And with the corrections made, we hope it's accepted for publication now. Thank you very much!

**'Break' and 'recombination' are nonstandard terminology for the phenomena the authors describe; for 'break' people usually say 'bifurcation', and for 'recombination', 'merging'. Actually, the 'break' that appears in the all-sky images looks to me like it could be the development of another bubble, or possibly the emergence of an arm on the side of their main bubble. These phenomena were discussed in detail, with simulations, by Huang et al. ( Huang, C.-S., J. M. Retterer, O. de La Beaujardiere, P. A.Roddy, D. E. Hunton, J. O. Ballenthin, and R. F. Pfaff (2012), Observations and simulations of formation of broad plasma depletions through merging process, J. Geophys. Res., 117, A02314, doi:10.1029/2011JA017084.)**

Reply: Thanks for your suggestions. After reading your comments and some references, we found it was inappropriate for using 'break' and 'recombination' in the manuscript. So, we used 'bifurcation' and 'merging' to replace 'break' and 'recombination', respectively. Besides, we also cited related reference in the revised manuscript.

**The variation of the zonal drift within plasma bubbles with both solar activity and**

**geomagnetic variations was discussed by Huang and Roddy (Huang, C.-S., and P. A.Roddy (2016), Effects of solar and geomagnetic activities on the zonal drift of equatorial plasma bubbles, J. Geophys. Res. Space Physics, 121, 628–637, doi:10.1002/2015JA021900.), which would be a useful reference here.**

Reply: Thanks for your valuable suggestions. After reading the reference, we find the viewpoint of this paper is consistent with our results. It is a very important reference for our manuscript. So, we have cited it in the revised manuscript.

**Finally, the presence of bubbles around sunrise was investigated thoroughly in the in-situ observations of the plasma density by the C/NOFS satellite, and those studies should be referenced here: Huang, C.-S., O. de La Beaujardiere, P. A. Roddy, D. E. Hunton, J. O. Ballenthin, and M. R. Hairston (2013),Long-lasting daytime equatorial plasma bubbles observed by the C/NOFS satellite,J. Geophys. Res. Space Physics,118,2398–2408, doi:10.1002/jgra.50252.**

Reply: Yes, we keep the same point as yours. Huang et al. (2013) reported the observations of long-lasting daytime EPBs with the C/NOFs satellite during a geomagnetic storm in which the EPBs were persistent from the post-midnight sector through the afternoon. So, we have cited it in the manuscript.

**Equatorial Plasma Bubbles Developing Around Sunrise Observed by an All-Sky Imager and GNSS Network during the Storm Time**

Kun Wu[1, 2], Jiyao Xu[1, 2], Xinan Yue[3, 2], Chao Xiong[4], Wenbin Wang[6], Wei Yuan[1, 2], Chi Wang[1, 2], Yajun Zhu[5], Ji Luo[1, 2]

[1]State Key Laboratory of Space Weather, National Space Science Center, Chinese Academy of Sciences, Beijing, China

[2]College of Earth and Planetary Sciences, University of Chinese Academy of Sciences, Beijing, China

[3]Key Laboratory of Earth and Planetary Physics, Institute of Geology and Geophysics, Chinese Academy of Sciences, Beijing, China

[4]GFZ German Research Centre for Geosciences, Telegrafenberg, 14473 Potsdam, Germany.

[5]Institute of Energy and Climate Research (IEK-7), Forschungszentrum Juelich GmbH, Juelich, Germany

[6]High Altitude Observatory, National Center for Atmospheric Research, Boulder, CO, USA

Correspondence to: jyxu@spaceweather.ac.cn

*Keywords:* Equatorial plasma bubble near sunrise, Spread-F, All-sky imager, GNSS network

**Manuscript-Tracked Changes**

**Abstract.**

A large number of studies have shown that equatorial plasma bubbles (EPBs) occur mainly after sunset, and they usually drift eastward. However, in this paper, an unusual EPB event was simultaneously observed by an all-sky imager and the Global Navigation Satellite Systems (GNSS) network in southern China, during the recovery phase of geomagnetic storm happened on 6-8 November 2015. Observations from both techniques show that the EPBs appeared near dawn. Interestingly, the observational results show that the EPBs continued to develop after sunrise, and disappeared about one hour after sunrise. The development stage of EPBs lasted for at least about 3 hours. To our knowledge, this is the first time that the evolution of EPBs developing around sunrise was observed by an all-sky imager and the GNSS network. Our observation showed that the EPBs drifted westward, which was different from the usually eastward drifts of post-sunset EPBs. The simulation from TIE-GCM model suggest that the westward drift of EPBs should be related to the enhanced westward winds at storm time. Besides, bifurcation and merging processes of EPBs were observed by the all-sky imager in the event. Associated with the development of EPBs, increasing in the ionospheric F region peak height was also observed near sunrise, and we suggest the enhance upward vertical plasma drift during geomagnetic storm plays a major role in triggering the EPBs near sunrise.

The texts of underline represent new contents or changed contents.

The texts of strikethrough indicate that contents have been deleted in the revised manuscript.

[revised manuscript text omitted]

Manuscript-Tracked Changes

Sheehan, R. E., &and Valladares, C. E. (2004). : Equatorial ionospheric zonal drift
model and vertical drift statistics from UHF scintillation measurements in South
America. Annales Geophysicae, Ann. Geophys., 22(9), 3177–3193.
https://doi.org/10.5194/angeo-22-3177-2004, 2004.

Scherliess, L., &and Fejer, B. G. (1997). : Storm time dependence of equatorial
disturbance dynamo zonal electric fields. Journal of Geophysical Research: Space
Physics, J. Geophys. Res., 102(A11), 24037–24046.
https://doi.org/10.1029/97ja02165, 1997.

Taylor, M. J., Eccles, J. V., Labelle, J., &and Sobral, J. H. A. (2013). : High resolution
oi (630 nm) image measurements of f‐region depletion drifts during the guará
campaign. Geophysical Research Letters, Geophys. Res. Lett., 24(13), 1699-
1702. https://doi.org/10.1029/97g101207, 2013.

Tulasi, R. S., Rama, R. P. V. S., Prasad, D. S. V. V. D., Niranjan, K., Gopi, K. S., &and
Sridharan, R., et al. (2008). : Local time dependent response of postsunset esf
during geomagnetic storms. Journal of Geophysical Research Space Physics, J.
Geophys. Res., 113, A07310. https://doi.org/10.1029/2007/JA012922, 2008.

Weber, E., Buchau, J., &and Moore, J. (1980). : Airborne studies of equatorial F layer
ionospheric irregularities. Journal of Geophysical Research, J. Geophys.
Res., 85(A9), 4631–4641. https://doi.org/10.1029/JA0 85iA09p04631, 1980.

Xiong, C., Park, J., Lühr, H., Stolle, C., &and Ma, S.Y. (2010). : Comparing plasma
bubble occurrence rates at CHAMP and GRACE altitudes during high and low
solar activity. Annales Geophysicae, Ann. Geophys., 28, 1647-1658.
https://doi.org/10.5194/angeo-28-1647-2010, 2010.

Xiong, C., Lühr, H., &and Fejer, B. G. (2015). : Global features of the disturbance
winds during storm time deduced from CHAMP observations. Journal of
Geophysical Research: Space Physics, J. Geophys. Res., 120, 5137–5150.
https://doi.org/10.1002/2015JA021302, 2015.

[revised manuscript text omitted]

Figure 1

[Figure]

Figure 2

[Figure]

Figure 3

[Figure]

Figure 4

[Figure]

Figure 5

[Figure]

[Figure]

Figure 6

[Figure]

[Figure]

Figure 7

[Figure]

[Figure]

Figure 8

[Figure]

Figure 9

[Figure]

Figure 10

[Figure]